# The Antibacterial Effect of PEGylated Carbosilane Dendrimers on *P. aeruginosa* Alone and in Combination with Phage-Derived Endolysin

**DOI:** 10.3390/ijms23031873

**Published:** 2022-02-07

**Authors:** Sara Quintana-Sanchez, Natalia Gómez-Casanova, Javier Sánchez-Nieves, Rafael Gómez, Jarosław Rachuna, Sławomir Wąsik, Jacek Semaniak, Barbara Maciejewska, Zuzanna Drulis-Kawa, Karol Ciepluch, F. Javier de la Mata, Michał Arabski

**Affiliations:** 1Department of Organic and Inorganic Chemistry, Research Institute in Chemistry “Andrés M. del Río” (IQAR), University of Alcalá, 28871 Alcalá de Henares, Spain; sara.quintana@edu.uah.es (S.Q.-S.); rafael.gomez@uah.es (R.G.); javier.delamata@uah.es (F.J.d.l.M.); 2Networking Research Center for Bioengineering, Biomaterials and Nanomedicine (CIBER-BBN), 28029 Madrid, Spain; 3Ramón y Cajal Institute of Health Research, IRYCIS, 28034 Madrid, Spain; 4Department of Biomedicine and Biotechnology, Faculty of Pharmacy, University of Alcalá, 28871 Alcalá de Henares, Spain; natalia.gomezc@uah.es; 5Division of Medical Biology, Jan Kochanowski University, 25-406 Kielce, Poland; jaroslaw.rachuna@gmail.com (J.R.); kciepluch@ujk.edu.pl (K.C.); 6Institute of Physics, Jan Kochanowski University, 25-406 Kielce, Poland; slawomir.wasik@ujk.edu.pl (S.W.); jacek.semaniak@ujk.edu.pl (J.S.); 7Department of Pathogen Biology and Immunology, University of Wroclaw, 51-148 Wroclaw, Poland; barbara.maciejewska@uwr.edu.pl (B.M.); zuzanna.drulis-kawa@uwr.edu.pl (Z.D.-K.)

**Keywords:** *Pseudomonas aeruginosa*, carbosilane dendrimers, polyethylene glycol, biofilm, endolysin, bacteriophages, laser interferometry

## Abstract

The search for new microbicide compounds is of an urgent need, especially against difficult-to-eradicate biofilm-forming bacteria. One attractive option is the application of cationic multivalent dendrimers as antibacterials and also as carriers of active molecules. These compounds require an adequate hydrophilic/hydrophobic structural balance to maximize the effect. Herein, we evaluated the antimicrobial activity of cationic carbosilane (CBS) dendrimers unmodified or modified with polyethylene glycol (PEG) units, against planktonic and biofilm-forming *P. aeruginosa* culture. Our study revealed that the presence of PEG destabilized the hydrophilic/hydrophobic balance but reduced the antibacterial activity measured by microbiological cultivation methods, laser interferometry and fluorescence microscopy. On the other hand, the activity can be improved by the combination of the CBS dendrimers with endolysin, a bacteriophage-encoded peptidoglycan hydrolase. This enzyme applied in the absence of the cationic CBS dendrimers is ineffective against Gram-negative bacteria because of the protective outer membrane shield. However, the endolysin—CBS dendrimer mixture enables the penetration through the membrane and then deterioration of the peptidoglycan layer, providing a synergic antimicrobial effect.

## 1. Introduction

*Pseudomonas aeruginosa* is one of the critical pathogens that the World Health Organization (WHO) includes in the list of priority pathogens with an urgent need for search and development of new antibiotics. Infections caused by this Gram-negative species are difficult to treat because of the multidrug resistance nature and are especially problematic in immunocompromised individuals or patients suffering from cystic fibrosis [1,2,3,4]. The bacterial outer membrane (OM) is the main protective shield of *P. aeruginosa* against toxic compounds and the immune response mechanisms. The specific composition of lipopolysaccharide (LPS) and a high content of cations (Mg^2+^) enables the formation of LPS–LPS bonds [5,6]. Therefore, *P. aeruginosa* can prevent the access of hydrophobic molecules to its interior. Moreover, the fast adaptation to unfavorable environmental conditions and biofilm formation hinders the activities of the immune system and the antibiotics [7]. Hence, there is currently a need to develop different therapeutic alternatives that are able to eradicate *P. aeruginosa* infections and fight the biofilm.

An important approach in antibacterial treatment is to design specific carriers transporting bioactive groups, protecting against degradation, increasing the circulation time and favoring solubility [8]. Among such carriers, dendrimers are great candidates due to their hyperbranched and well-defined structure, monodispersity and nanometric size [9,10]. Nowadays, different dendritic systems are being developed with a variety of frameworks [11,12,13], such as PAMAM (poly(amidoamine)), PPI (poly(propylene imine)), carbosilane (CBS), phosphorus and polyether or polyester dendrimers.

The antibacterial activity of dendritic systems containing specific moieties such as ammonium groups (QAS) has been widely studied [14,15,16,17,18]. These polycationic dendrimers interact with the OM of Gram-negative bacteria by electrostatic interaction with the phosphate groups of LPS. The OM is disrupted by the shifting of divalent cations (Mg^2+^, Ca^2+^), and hence, dendrimers can pass through the cell membrane into the bacterial cell [6,19]. This penetration requires an adequate hydrophobic/hydrophilic balance of the dendrimer structure. In the case of CBS dendrimers, the skeleton is hydrophobic while the outermost ammonium groups form the hydrophilic part of the macromolecule, providing a good biocide activity against both planktonic and sessile cells [20,21,22].

On the other hand, polyethylene glycol (PEG) is a hydrophilic polymer that is not toxic to eukaryotic cells, as well as being non-immunogenic and non-antigenic [23]. Thus, PEGylation is a procedure favoring solubility in water reducing intramolecular aggregation and toxicity and increasing the circulation time of potential drug carriers, since the presence of PEG inhibits the opsonization and phagocytosis [24]. It was also reported that PEG allows the diffusion process through mucous fluids such as biofilms. Therefore, an improved therapeutic efficacy against biofilm-inhabiting bacteria can be expected for biocide systems modified with PEG [25]. Previous studies on dendrimer PEGylation and their antibacterial activity showed that these systems required the achievement of a proper balance between cationic groups and PEG attached to obtain an enhanced antimicrobial effect [15,26].

Polycationic dendritic systems serving as bacterial OM destabilizers can also be used in combination with other antimicrobials. It is especially beneficial in the treatment of Gram-negative bacteria when combined with peptidoglycan (PG)-degrading enzymes such as endolysin [22,26]. Since the OM efficiently protects bacterial PG from externally applied agents, most of the endolysins that are applied alone exhibit significant activity only against Gram-positive bacteria.

This work aims to evaluate the influence of PEG modification on the antibacterial activity of cationic dendrimers. First, the CBS dendrimers were functionalized with ammonium groups (-NMe_3_^+^) alone or with PEG (molecular weight of 800 Dalton (PEG800) or 2K Dalton (PEG2K)) and then tested against planktonic and biofilm-forming *P. aeruginosa* cells. Then, the influence of the PEG ligands was analyzed by laser interferometry and fluorescence microscopy to determine the ability of the dendrimers to diffuse through a model biofilm. Finally, the antibacterial effect of cationic CBS dendrimer/endolysin combined treatment was investigated. This study provided a better understanding of the cationic dendritic systems’ activity mechanism while pegylated or without PEGylation, hence, improving the design of future pharmaceutical strategies.

## 2. Results

### 2.1. Synthesis and Characterization of Cationic CBS Dendrimers Homo- and Heterofunctionalized with Ammonium Groups and Polyethylene Glycol

Previously, we observed that cationic CBS dendrimers of low generation also containing a sulfur atom at the periphery showed no bactericidal activity, whereas analogous dendrimers of higher generation were rather active. This difference was attributed to the lack of hydrophobicity in the low-generation dendrimer [20]. Herein, we increased the hydrophobic framework of a cationic CBS dendrimer of low generation intending to improve its antibacterial properties but without increasing the number of cationic groups.

First, the precursor dendrimer with four vinyl groups was obtained from tetravinylsilane by hydrosilylation and alkenylation, following the same procedure to grow CBS dendrimers (see experimental methodology), but in this case, instead of increasing the generation, only the branches were enlarged. The surface of this dendrimer was modified with the corresponding thiol derivative by thiol-ene reaction, under ultraviolet light conditions [20]. Then, dimethylammonium groups were neutralized with sodium hydrocarbonate. Finally, the quaternization of dimethylamine groups was carried out in presence of an excess of iodomethane (Figure 1). All reactions have been monitored by nuclear magnetic resonance (NMR) spectroscopy and the final product (**4**, Figure 2) was characterized by ^1^H-NMR, ^13^C-NMR, elemental analysis and mass spectrometry.

In addition, to introduce a biocompatible group, i.e., PEG, in the surface of the dendrimers [24,25] and to determine the influence of the PEG’s length in the bactericidal properties of these cationic CBS dendrimers, two heterofunctionalized CBS dendrimers with cationic trimethylammonium groups and a PEG moiety (PEG-800 (**7**) and PEG-2K (**10**)) were prepared (Figure 2).

These new dendrimers were obtained by also employing thiol-ene. In a first step, the thiol-PEG reagent was added and, after checking by NMR the modification of the dendrimer surface with this ligand, the complete transformation of the surface was accomplished by the addition of 2-(dimethylamino)ethanethiol hydrochloride over the previous solution (Figure 1). Compounds **7** (PEG-800) and **10** (PEG-2K) were characterized by NMR spectroscopy and elemental analysis.

Finally, for all the cationic systems, we replaced the iodide anion (I^−^) with a chloride anion (Cl^-^) using an excess of the Amberlite IRA-402, Cl^−^ form. Thanks to this anion exchange, we increased the solubility of cationic dendrimers.

### 2.2. Antibacterial Activity of Dendrimers against P. aeruginosa

The antibacterial activity of homofunctionalized (**4**) and heterofunctionalized (**7**, **10**) cationic CBS dendrimers was studied in *P. aeruginosa* planktonic cells and biofilm. Our data showed that treatment with compound **4** (minimum inhibitory concentration (MIC) and minimum bactericidal concentration (MBC) of 32 mg/L) was more effective than that with **7** and **10** dendrimers (MIC and MBC of 256 for both compounds) on planktonic cells (Table 1). Thus, PEG heterofunctionalization of dendritic systems decreased the antibacterial activity. A similar effect was observed for the biofilm-forming culture as well as its degradation but at a higher concentration of dendrimers.

The cell morphology changes after incubation with tested dendrimers at concentrations at and below their MBCs (Figure 3) were examined using scanning electron microscopy (SEM). Cells showed an absence of smooth walls on their surface, and we even found flattened cells (Figure 3, white arrows). The altered morphology of elongated bacterial planktonic cells was observed after treatment with all tested dendrimers. The cell density in the presence of compound **4** was the lowest compared to the control and pegylated **7** and **10** samples.

### 2.3. Diffusion Properties and Antibiofilm Activity of Dendrimers on P. aeruginosa PAO1 Biofilm

For laser interferometry and fluorescence study, the most active CBS dendrimers, one functionalized only with ammonium groups (**4**) and one with PEG800 and ammonium groups (**7**), were selected. Compound **4**, as a fully cationic dendrimer, can interact with the negatively charged biofilm thereby becoming trapped in the matrix, in contrast to the PEGylated dendrimer version **7**. Thus, no diffusion of compound **4** through the biofilm layer was observed by laser interferometry, whereas dendrimer **7** was able to efficiently pass through the mature PAO1 biofilm (9.83 × 10^−9^ mol after 40 min) (Figure 4A). The above results corresponded with the biofilm-forming bacteria eradication measured by fluorescence microscopy (Figure 4B). The live/dead cell staining showed that the cationic CBS dendrimer (**4**) has much higher anti-biofilm properties compared to dendrimer functionalized with PEG800 (**7**). Nevertheless, both types of dendrimers were able to kill bacteria compared to the non-treated control.

### 2.4. Antibacterial Activity of Dendrimer Combined with Endolysin against P. aeruginosa PAO1

Endolysins are a type of enzyme that can be a biocide against both Gram-positive and Gram-negative bacteria [27]. However, with some single exceptions of endolysins reviewed by Gerstmans et al. that permeate the OM barrier per se, the exogenous application of native phage endolysins on Gram-negatives is rather ineffective [28]. Nevertheless, our previous studies on cationic dendrimers with different skeleton nature showed their positive impact on the improvement of endolysin antibacterial activity [22,26,29]. In this work, KP27 endolysin with the activity of l-alanyl-d-glutamate endopeptidase was mixed with the CBS dendrimer functionalized with ammonium groups (NMe_3_^+^) alone (**4**) or with PEG (PEG800 (**7**), PEG2K (**10**)) to verify a possible synergic effect (Figure 5). The antibacterial properties were evaluated on PAO1 wild-type and its ∆*wbpL* knock-out mutant lacking lipopolysaccharide O-chain.

It turned out that PEGylation of CBS dendrimers alone and in combination with phage-derived endolysin reduced their antibacterial properties against *P. aeruginosa* PAO1 wild-type and its ∆*wbpL* mutant. The CBS dendrimer functionalized with ammonium groups (-NMe_3_^+^) (**4**) at the concentration of 56 mg/L showed a statistically significant decreased optical density of PAO1 cells (*p* = 0.031 compared to untreated control; One-Way ANOVA test). The antibacterial effect was higher for the combination with endolysin compared to dendrimer alone (*p* = 0.042). A similar effect of compound **4** was observed for the ∆*wbpL* mutant (*p* = 0.016) but the antagonistic effect of endolysin was not observed. It indicates the role of antigen O from LPS structure on the antibacterial effect of dendrimer in combination with endolysin.

## 3. Discussion

The first aim of this study was to evaluate the antibacterial activity of CBS dendrimers functionalized with ammonium groups (-NMe_3_^+^) alone or with PEG (PEG800, PEG2K). The dendrimers were tested against planktonic (alone or in combination with endolysin) and biofilm-forming *P. aeruginosa* cells.

Results in planktonic bacteria showed that PEGylation of the dendrimers (**7** and **10**; MBC at 256 mg/L) decreases the antibacterial activities giving the MBC of dendrimers eight times higher than that of homofunctionalized CBS dendrimers (**4**; MBC at 32 mg/L). The tendency to diminish the bactericide effect when dendrimers are PEGylated might be associated with the decrease in the number of cationic ammonium groups and the shielding of the positive charges by the PEG chains. These facts can be correlated with the decrease of the electrostatic interactions between dendrimers and the negatively-charged bacterial surface. It might promote nanopores formation [30]. The effect of PEGylation on the reducing antibacterial properties of dendrimers was previously observed in the case of the G3 and G5 PAMAM dendrimers. It was observed for a higher number of PEG ligands and, consequently, a lower number of ammonium groups [31]. On the other hand, metallic nanoparticles functionalized with cationic CBS dendrons (different generations) and PEG chains, showed the same trend, probably because the PEG and the dendrons had similar size and the PEG chains could not shield the positive charge of ammonium groups [32].

In addition, the presence of long PEG chains also reduced the antibiofilm properties of CBS dendrimers. Regarding the biofilm-damaging (minimum biofilm damaging concentration, MBDC) and -eradicating concentrations (minimum biofilm eradication concentration, MBEC) against an established *P. aeruginosa* biofilm, dendrimer **4** was the best biocide with an MBDC of 128 mg/L and MBEC of 512 mg/L. The lower antibacterial activity of PEGylated CBS dendrimers could be related to the effectiveness of their diffusion through a mature *P. aeruginosa* biofilm. The results showed that dendrimer **4** interacted with sessile cells in contrast to PEGylated compound **7**, in which PEGylation reduced the positive charge and the biocidal activity.

The molecular weights of these three dendrimers are notably different. For this reason, we compared the activity taking into account molar concentration (see Appendix A). UnPEGylated dendrimer **4** was the most active with this consideration also. However, among both PEGylated dendrimers, dendrimer **10**, with the longest PEG chain, proved to be slightly more active per molecule than dendrimer **7**, with the shortest PEG chain. This indicates that the long PEG chain exerts a positive action regarding antibacterial activity when compared with the short one.

On the other hand, images obtained using SEM allowed us to observe the formation of long filaments, especially after treatment with unPEGylated dendrimer **4** (16 mg/L) and PEGylated dendrimer **10** (256 mg/L). This result suggests that treatment with these dendrimers affected the cell septation during the cell division process. The filamentation commonly originates as a consequence of DNA damage or stress conditions [33]. In particular, the inhibition of PBP-3 activity is related to filamentous bacterial morphology [34]. The few cells we found at the MBC concentration were damaged. The absence of viability of these cells was corroborated by the drop plate method, suggesting that the cells could have DNA damage [33].

The next aim of this study was to evaluate the biocide effect of endolysin in combination with tested dendrimers against *P. aeruginosa* as the representative of Gram-negative bacteria. While the endolysins of Gram-positive specific phages show high bactericidal activity, the lytic activity of endolysins against Gram-negatives is limited by the presence of a protective OM. Although some endolysins that are able to pass the OM barrier and act exogenously against Gram-negative bacteria have already been described, most of them must be assisted by membrane destabilizers. There are two general approaches to support the lytic activity of endolysins.

The first approach relies on the fusion of endolysin with cationic peptides (mainly by genetic engineering) and is based on observation of endolysins, which have an intrinsic antibacterial activity against Gram-negatives. This natural activity depends on the presence of short but highly positively charged domains, which have been proved to be strategic for the antibacterial function of *Thermus* phage Ts2631 endolysin [35], *Acinetobacter baumannii* phage lysin PlyF307 [36] and LysC lysin encoded by *Clostridium intestinale* [37].

The second approach relies on a combination of endolysin with some OM-permeabilizing agents such as organic acids, liposomes, antibiotics or dendrimers [38,39,40]. The enhanced effect of endolysin mixed with unPEGylated dendrimer against *P. aeruginosa* cells might indicate the synergistic mechanism, in which positively charged dendrimers enhanced the OM’s permeability to peptidoglycan-degrading enzyme. This effect confronted the results observed for endolysin combination with amine PPI dendrimers, where no improvement of the antibacterial activity was seen against the PAO1 strain [26]. However, when endolysin was combined with AgNP modified with PEG and cationic CBS dendrons, the bactericidal activity increased notably. Therefore, cationic CBS systems targeting lipopolysaccharides might be useful for OM barrier disruption, thus endolysin to penetrate to the periplasmic space and reach the peptidoglycan [22]. It is also worth noting that in the case of the PEG800 dendrimer, the antimicrobial effect is weaker for the ∆wbpL mutant compared to that for the wild-type strain. This suggests that dendrimers must first interact with the O-chain (the carbohydrate moiety) of the LPS and then act on the membrane. Our previous study on dendronized silver nanoparticles indisputably shows that PEGylated dendronized AgNPs can complex with LPS, while the LPS–non-PEGylated NP complex formation effect is not favored due to the tendency of non-PEGylated dendronized AgNPs to aggregate [22].

This study confirmed an idea for an antibacterial system consisting of cationic dendrimers and lytic proteins such as endolysin for the design of future pharmaceutical strategies.

## 4. Materials and Methods

### 4.1. Synthesis of Cationic CBS Dendrimers

All synthetic protocols and characterization and purification processes for each dendrimer are described below. The reagents, tetravinylsilane, chlorodimethylsilane, Karstedt’s Pt catalyst, 2-(dimethylamino)ethanethiol hydrochloride, poly(ethylene glycol) methyl ether thiol (Mw 800 and 2000), 2,2-dimethoxy-2-phenylacetophenone (DMPA), NaH(CO_3_) and MeI, were obtained from commercial sources and used as received. An HPK 125 W Mercury Lamp from Heraeus Noblelight (UV Consulting Peschl, Valencia, Spain) with maximum energy at 365 nm was employed to carry out thiol-ene reactions. NMR spectra were recorded on a Bruker Neo400-(Bruker, Madrid, Spain) (400.13 (^1^H), 100.60 (^13^C) MHz). Chemical shifts (δ) were given in ppm. ^1^H and ^13^C resonances were measured relative to solvent peaks considering TMS = 0 ppm. Elemental analyses were performed on a LECO CHNS-932 (Leco, Madrid, Spain).The NMR spectra are presented in Appendix A).

**G_0_Si(SiMe_2_V)_4_ (1).** Compound **1** was obtained by hydrosilylation and alkylation reactions. First, an excess of chlorodimethylsilane (3.56 mL, 0.031 mol) was added to tetravinylsilane (0.68 mL, 0.0039 mol) and a drop of Karsted catalyst. The reaction mixture was stirred for 30 min at 5 °C. Then, the mixture was warmed at 60 °C for 16 h, at which time the reaction was stopped. ^1^H-NMR signals of vinyl groups had disappeared completely, as expected. Subsequently, the alkylation reaction was carried out, adding, drop by drop, vinyl magnesium bromide (17.1 mL, 0.017 mol) to a diethyl ether solution of the product previously obtained. After 24 h stirring at room temperature, the reaction was finished. The crude product was purified through Et_2_O/H_2_O (NaCl) extraction, with Compound 1 being separated with the organic phase. The organic phase was dried with MgSO_4_ and filtered with active carbon to remove the Karsted catalyst. Then, the solvent was removed by rotatory evaporation. A yellow oil was obtained as **1** (1.3 g). Data for **1**: ^1^H-NMR (CDCl_3_): δ = 0.07 (s, 24H, Si*Me*_2_(CHCH_2_)), 0.43 (m, 16H, SiC*H*_2_CH_2_Si, SiCH_2_C*H*_2_Si), 5.64–6.20 (m, 12H, SiC*H*CH_2_, SiCHC*H*_2_). ^13^C{H} NMR: δ = −3.41 (Si*Me*_2_(CHCH_2_)), 3.15 (Si*C*H_2_CH2Si), 7.90 (Si*C*H_2_*C*H2Si), 131.57 (SiCH*C*H_2_), 139.18 (Si*C*HCH_2_). C_24_H_52_Si_5_ (481.1 g/mol). Calc.: C, 59.92; H, 10.89; Obt.: C, 59.05; H, 9.85.

**G_0_Si(SiMe_2_-NMe_2_HCl)_4_ (2).** Compound **1** (0.852 g, 1.8 mmol, 1 equiv.) was dissolved in THF/MeOH (1:2) and mixed with 2-(dimethylamino)ethanethiol hydrochloride (1.05 g, 7.1 mmol, 4 equiv.) and DMPA (5%). The reaction mixture was deoxygenated and stirred under ultraviolet light. After 4 h, the reaction was finished and then measured by ^1^H-NMR. Then, solvents were removed by rotatory evaporation. The dendrimer was redissolved in distilled water and filtered with a 0.22 μm syringe filter to remove the DMPA. Compound **2** was obtained as an orange solid product (0.856 g). Data for **2**: ^1^H-NMR (CDCl_3_): δ = −0.02 (s, 24H, Si*Me*_2_(CHCH_2_)), 0.35 (m, 16H, SiC*H*_2_CH_2_Si, SiCH_2_C*H*_2_Si), 0.85 (m, 8H, *SiCH*_2_CH_2_*S*), 2.60 (m, 8H, SiCH_2_C*H*_2_S), 2.87 (s, 24H, NH*Me*^+^_2_), 2.98 (m, 8H, SC*H*_2_CH_2_N), 3.23 (m, 8H, SCH_2_C*H*_2_N). ^13^C{H} NMR: δ = −3.55 (Si*Me*_2_), 2.87 (Si*C*H_2_CH_2_Si), 7.54 (SiCH_2_CH_2_Si), 15.87 (SiCH_2_CH_2_S), 26.23 (SiCH_2_CH_2_S), 28.74 (SCH_2_CH_2_N), 43.34 (NH*Me*_2_^+^), 57.72 (SCH_2_*C*H_2_N).

**G_0_Si(SiMe_2_-NMe_2_)_4_ (3)**. The neutralization of ammonium groups of dendrimer **2** (0.85 g, 0.81 mmol) was carried out with an excess of sodium hydrogen carbonate. The reagents of the reaction were dissolved in distilled water and stirred for 1 h. The reaction mixture was extracted with CH_2_Cl_2_/H_2_O, in triplicate. The organic phase was extracted with the dendrimer neutralized and dried over Na_2_SO_4_. Compound **3** was purified after solvent removal under vacuum conditions (0.76 g). Data for **3**: ^1^H-NMR (CDCl_3_): δ = −0.10 (s, 24H, Si*Me*_2_), 0.26 (m, 16H, SiC*H*_2_CH_2_Si, SiCH_2_C*H*_2_Si), 0.79 (m, 8H, SiC*H*_2_CH_2_S), 2.15 (s, 24H, N*Me*_2_), 2.39–2.54 (m, 24H, SiCH_2_C*H*_2_S, SC*H*_2_CH_2_N, SCH_2_C*H*_2_N). ^13^C{H} NMR: δ= −3.80 (Si*Me*_2_), 2.75 (Si*C*H_2_CH_2_Si), 7.41 (SiCH_2_*C*H_2_Si), 15.69 (Si*C*H_2_CH_2_S), 28.00 (SiCH_2_*C*H_2_S), 29.86 (S*C*H_2_CH_2_N), 45.56 (N*Me*_2_), 59.48 (SCH_2_*C*H_2_N). C_40_H_96_N_4_S_4_Si_5_ (901.90 g/mol). Calc.: C, 53.27; H, 10.73; N, 6.21; S, 14.22; Obt.: C, 53.56; H, 10.39; N, 6.268; S, 14.26.

**G_0_Si(SiMe_2_-NMe_3_Cl)_4_ (4).** Dendrimer **4** was prepared from compound **3**. An excess of iodomethane was mixed with compound **3** (0.41 g, 0.45 mmol) previously dissolved in dry THF. The sample was stirred for 16 h at room temperature. Next, excess MeI was removed under vacuum. The product was redissolved in distilled water and the I^-^ anion was replaced by the Cl^-^ anion with Amberlite IRA-402, Cl^-^ form. Compound **4** was obtained as a yellow solid product (0.49 g). Data for **4**: ^1^H-NMR (D_2_O): δ = −0.02 (s, 24H, Si*Me*_2_), 0.38 (m, 16H, SiC*H*_2_CH_2_Si, SiCH_2_C*H*_2_Si), 0.82 (m, 8H, SiC*H*_2_CH_2_S), 2.64 (m, 8H, SiCH_2_C*H*_2_S), 2.90 (m, 8H, SC*H*_2_CH_2_N), 3.09 (s, 36H, N*Me_3_*^+^), 3.47 (m, 8H, SCH_2_C*H*_2_N). ^13^C{H} NMR: δ = −3.50 (Si*Me*_2_), 3.02 (Si*C*H_2_CH_2_Si), 7.61 (SiCH_2_*C*H_2_Si), 15.62 (Si*C*H_2_CH_2_S), 24.57 (SiCH_2_*C*H_2_S), 28.21 (S*C*H_2_CH_2_N), 53.42 (N*Me_3_*^+^), 65.88 (SCH_2_*C*H_2_N). C_44_H_108_I_4_N_4_S_4_Si_5_ (1470 g/mol). Calc.: C, 35.96; H, 7.41; N, 3.81; S, 8.73. Obt.: C, 35.46; H, 7.22; N, 3.84; S, 8.23.

**G_0_Si(SiMe_2_-PEG800)(SiMe_2_-NMe_2_HCl)_3_ (5).** Compound **5** was prepared through a combination of click thiol-ene reactions. Polyethylene glycol (Mw 800, 0.329 g, 0.41 mmol, 0.9 equiv.) and DMPA (5%) were added to a THF/MeOH (1:2) solution of 1 (0.220 g, 0.46 mmol, 1 equiv.). The reaction mixture was deoxygenated and stirred under ultraviolet light for 30 min. Then, 2-(dimethylamino)ethanethiol hydrochloride (0.194 g, 1.4 mmol, 3 equiv.) and DMPA (5%) were added, and the reaction mixture was irradiated for another 4 h. The reaction was monitored by ^1^H-RMN. Afterward, solvents were removed by rotary evaporation. The crude product was purified by filtering the dendrimer redissolved in distilled water with a 0.22 μm syringe filter to remove DMPA, and then by dialysis (membrane of 100–500 Da). Dendrimer **5** was obtained as an orange oil (0.392 g). Data for **5**: ^1^H-NMR (D_2_O): δ = −0.07 (s, 24H, Si*Me*_2_), 0.33 (m, 16H, SiC*H*_2_CH_2_Si, SiCH_2_C*H*_2_Si), 0.79 (m, 8H, SiC*H*_2_CH_2_S), 2.55 (m, 10H, SiCH_2_C*H*_2_S, SiC*H*_2_CH_2_O), 2.69 (s, 18H, NH*Me*_2_^+^), 2.76 (m, 6H, SC*H*_2_CH_2_N), 3.10 (m, 6H, SCH_2_C*H*_2_N), 3.25 (s, 3H, *OCH_3_*), 3.57 (m, ~70H, SCH_2_C*H*_2_O, OC*H*_2_C*H*_2_O). ^13^C{H} NMR: δ = −4.01 (Si*Me*_2_), 2.49 (Si*C*H_2_CH_2_Si), 7.04 (SiCH_2_*C*H_2_Si), 15.06 (Si*C*H_2_CH_2_S), 25.94 (SiCH_2_*C*H_2_S), 27.31 (S*C*H_2_CH_2_N), 30.84 (S*C*H_2_CH_2_O), 42.97 (NH*Me*_2_^+^), 56.85 (SCH_2_*C*H_2_N), 58.06 (OC*H_3_*), 69.57 (O*C*H_2_*C*H_2_O), 70.98 (SCH_2_*C*H_2_O).

**G_0_Si(SiMe_2_-PEG800)(SiMe_2_-NMe_2_)_3_ (6).** Compound **6** was prepared by the neutralization of compound **5** (0.353 g, 0.21 mmol) with an excess of sodium hydroxide (0.041 g, 1.03 mmol) in a distilled water solution. The mixture was extracted with H_2_O/CH_2_Cl_2_. Compound **6** was dissolved and separated with the organic phase. The aqueous phase was extracted twice with CH_2_Cl_2_. The organic phase was dried over Na_2_SO_4_ and CH_2_Cl_2_ was removed by rotary evaporation to give **6** as an orange oil (0.258 g). Data for **6**: ^1^H-NMR (CDCl_3_): δ = −0.01 (s, 24H, Si*Me*_2_), 0.37 (m, 16H, SiC*H*_2_CH_2_Si, SiCH_2_C*H*_2_Si), 0.87 (m, 8H, SiC*H*_2_CH_2_S), 2.24 (s, 18H, N*Me*_2_), 2.51–2.71 (m, 16H, SiCH_2_C*H*_2_S, SC*H*_2_CH_2_N, SC*H*_2_CH_2_O), 3.36 (s, 3H, OC*H_3_*), 3.62 (m, ~70H, OC*H*_2_CH_2_O, SCH_2_C*H*_2_O). ^13^C{H} NMR: δ = −3.58 (Si*Me*_2_), 2.97 (Si*C*H_2_CH_2_Si), 7.61 (SiCH_2_*C*H_2_Si), 15.87 (Si*C*H_2_CH_2_S), 28.14 (SiCH_2_*C*H_2_S), 29.95 (S*C*H_2_CH_2_N), 31.50 (S*C*H_2_CH_2_O), 45.59 (N*Me*_2_), 59.18 (O*C*H_3_), 59.49 (SCH_2_*C*H_2_N), 70.71 (O*C*H_2_*C*H_2_O), 72.07 (SCH_2_*C*H_2_O). C_71_H_157_N_3_O_17_S_4_Si_5_ (1596.7 g/mol). Calc.: C, 53.54; H, 9.91; N, 2.57; S, 7.83. Obt.: C, 53.74; H, 9.455; N, 2.851; S, 7.460.

**G_0_Si(SiMe_2_-PEG800)(SiMe_2_-NMe_3_Cl)_3_ (7).** Compound **6** was quaternized following the same method as for compound **4**. Excess of MeI was mixed with compound **6** (0.258 g, 0.81 mmol) previously dissolved in dry THF. The pure product was obtained as an orange solid sample (0.145 g). Data for **7**: ^1^H-NMR (D_2_O): δ = −0.04 (s, 24H, SiMe_2_), 0.35 (m, 16H, SiC*H*_2_CH_2_Si, SiCH_2_C*H*_2_Si), 0.80 (m, 8H, SiC*H*_2_CH_2_S), 2.53 (m, 2H, SiC*H*_2_CH_2_O), 2.63 (m, 8H, SiCH_2_C*H*_2_S,), 2.89 (m, 6H, SC*H*_2_CH_2_N), 3.08 (s, 27H, N*Me_3_*^+^), 3.27 (s, 3H, OC*H_3_*), 3.47 (m, 6H, SCH_2_C*H*_2_N), 3.51 (m, OC*H*_2_CH_2_O, SCH_2_C*H*_2_O). ^13^C{H} NMR: δ = −4.12 (Si*Me*_2_), 2.33 (Si*C*H_2_CH_2_Si), 7.01 (SiCH_2_*C*H_2_Si), 15.13 (Si*C*H_2_CH_2_S), 24.11 (S*C*H_2_CH_2_N), 27.66 (SiCH_2_*C*H_2_S, S*C*H_2_CH_2_O), 52.98 (N*Me_3_*^+^), 57.98 (O*CH_3_*), 65.40 (SCH_2_*C*H_2_N), 69.60 (O*C*H_2_*C*H_2_O), 70.89 (SCH_2_*C*H_2_O). C_74_H_166_I_4_N_3_O_17_S_4_Si_5_ (2022.52 g/mol). Calc.: C, 44.01; H, 8.29; N, 2.08; S, 6.35. Obt.: C, 39.75; H, 8.04; N, 2.05; S 5.40.

**G_0_Si(SiMe_2_-PEG2K)(SiMe_2_-NMe_2_HCl)_3_ (8)**. A THF/MeOH (1:2) solution of **1** (0.100 g, 0.21 mmol, 1 equiv.) was mixed; first, with polyethylene glycol (Mw 2000, 0.374 g, 0.19 mmol, 0.9 equiv.) and DMPA (5%), and afterward, 2-(dimethylamino)ethanethiol hydrochloride (0.093 g, 0.62 mmol, 3 equiv.) and DMPA (5%) were added. Data for **8**: ^1^H-NMR (CDCl_3_): δ = −0.08 (s, 24H, SiMe_2_), 0.32 (m, 16H, SiC*H*_2_CH_2_Si, SiCH_2_C*H*_2_Si), 0.77 (m, 8H, SiC*H*_2_CH_2_S), 2.56 (m, 10H, SiCH_2_C*H*_2_S, SiC*H*_2_CH_2_O), 2.79 (m, 24H, NH*Me*_2_^+^, SC*H*_2_CH_2_N), 3.20–3.23 (m, 9H, SCH_2_C*H*_2_N, OCH_3_), 3.48 (m, 2H, SCH_2_C*H*_2_O), 3.55 (m, ~174H, OC*H*_2_C*H*_2_O). ^13^C{H} NMR: δ = −4.01 (Si*Me*_2_), 2.46 (Si*C*H_2_CH_2_Si), 7.12 (SiCH_2_*C*H_2_Si), 15.04 (Si*C*H_2_CH_2_S), 25.58 (SiCH_2_*C*H_2_S), 27.33 (S*C*H_2_CH_2_N), 30.96 (S*C*H_2_CH_2_O), 42.75 (NH*Me*_2_^+^), 56.51 (SCH_2_*C*H_2_N), 58.05 (OC*H_3_*), 69.57 (O*C*H_2_*C*H_2_O), 70.98 (SCH_2_*C*H_2_O).

**G_0_Si(SiMe_2_-PEG2K)(SiMe_2_-NMe_2_)_3_ (9).** In this case, compound **8** (0.206 g, 0.074 mmol, 1 equiv.) was neutralized with an excess of sodium hydroxide (5 equiv.) in a distilled water solution. Product **9** was obtained as a yellow oil (0.104 g). Data for **9**: ^1^H-NMR (CDCl_3_): δ = −0.04 (s, 24H, Si*Me*_2_), 0.33 (m, 16H, SiC*H*_2_CH_2_Si, SiCH_2_C*H*_2_Si), 0.84 (m, 8H, SiC*H*_2_CH_2_S), 2.22 (s, 18H, N*Me*_2_), 2.43–2.63 (m, 20H, SiCH_2_C*H*_2_S, SC*H*_2_CH_2_N, SCH_2_C*H*_2_N), 2.68 (m, 2H, SiC*H*_2_CH_2_O), 3.33 (s, 3H, OC*H_3_*), 3.60 (m, ~174H, OCH_2_C*H*_2_O, SC*H*_2_CH_2_O). ^13^C{H} NMR: δ = −3.75 (Si*Me*_2_), 2.82 (Si*C*H_2_CH_2_Si), 7.49 (SiCH_2_*C*H_2_Si), 15.77 (Si*C*H_2_CH_2_S), 28.09 (SiCH_2_*C*H_2_S), 29.91 (S*C*H_2_CH_2_N), 31.45 (S*C*H_2_CH_2_O), 45.60 (N*Me*_2_), 59.26 (O*C*H_3_), 59.54 (SCH_2_*C*H_2_N), 70.80 (O*C*H_2_*C*H_2_O), 72.16 (SCH_2_*C*H_2_O). C_125_H_265_N_3_O_44_S_4_Si_5_ (2796.7 g/mol). Calc.: C, 53.95; H, 9.60; N, 1.51; S, 4.61. Obt.: C, 53.92; H, 9.563; N, 2.226; S, 4.610.

**G_0_Si(SiMe_2_-PEG2K)(SiMe_2_-NMe_3_Cl)_3_ (10**). Excess of MeI was mixed with compound **9** (0.206 g, 0.074 mmol) previously dissolved in dry THF. A yellow solid product was obtained (0.218 g). Data for **10**: ^1^H-NMR (D_2_O): δ = −0.05 (bs (broad singlet), 24H, SiMe_2_), 0.35 (m, 16H, SiC*H*_2_CH_2_Si, SiCH_2_C*H*_2_Si), 0.80 (m, 8H, SiC*H*_2_CH_2_S), 2.52–2.62 (m, 10H, SiCH_2_C*H*_2_S, SiC*H*_2_CH_2_O), 2.88 (m, 8H, SC*H*_2_CH_2_N), 3.07 (s, 27H, N*Me_3_*^+^), 3.26 (s, 3H, OC*H_3_*), 3.46 (m, 6H, SCH_2_C*H*_2_N), 3.50 (m, ~174H, OC*H*_2_C*H*_2_O), 3.58 (m, 2H, SCH_2_C*H*_2_O). ^13^C{H} NMR: δ = −3.95 (Si*Me*_2_), 1.47 (Si*C*H_2_CH_2_Si), 6.70 (SiCH_2_*C*H_2_Si), 14.69 (Si*C*H_2_CH_2_S), 23.80 (S*C*H_2_CH_2_N), 27.52 (SiCH_2_*C*H_2_S), 30.50 (S*C*H_2_CH_2_O), 52.85 (N*Me_3_*^+^), 57.94 (O*CH_3_*), 65.47 (SCH_2_*C*H_2_N), 69.51 (SCH_2_*C*H_2_O, O*C*H_2_*C*H_2_O). C_128_H_274_I_4_N_3_O_44_S_4_Si_5_ (3222.8 g/mol). Calc.: C, 47.91; H, 8.61; N, 1.31; S, 4.00. Obt.: C, 47.65; H, 7.46; N, 1.57; S, 3.93.

### 4.2. Bacterial Strains

The antibacterial activity of biocides was tested using *P. aeruginosa* wild-type strain from the Colección Española de Cultivos Tipo (CECT) 108 of the Department of Biotechnology and Biomedicine, Universidad de Alcalá (Spain). Additionally, the effect of combined cationic CBS dendrimers with endolysin was tested on *P. aeruginosa* ATCC15692 (PAO1) wild-type strain obtained from the Department of Biochemistry and Genetics, Jan Kochanowski University (Kielce, Poland) and its knock-out ∆*wbpL* mutant deficient in the lipopolysaccharide (LPS) biosynthesis (lack of A-band and B-band in the O-antigen) provided by Andrew M. Kropinski from the Laboratory of Foodborne Zoonoses, Guelph, ON, Canada.

### 4.3. Analysis of Antibacterial Properties of Dendrimers against Planktonic Cells

The antibacterial activity on planktonic *P. aeruginosa* PAO1 cells was studied following the ISO 20776-1:2006 protocol. Thirteen concentrations of each dendrimer, in the range of 0.125 to 512 mg/L, were tested in triplicate in 96-well microtiter plates. Control wells were included in all experiments: *P. aeruginosa* culture in tryptic soy broth (TSB) medium without biocide (positive control), TSB medium containing the biocide (negative control) and wells containing the TSB medium only (negative control). The final volume of tested samples was 200 μL containing 5 × 10^5^ CFU/mL of bacterial culture (CFU, colony-forming units). Samples were incubated for 20 h at 37 °C, and the absorbance at 630 nm was measured by the microplate reader ELX808iu (BioTek Instruments, Winooski, VT, USA) to determine the minimal inhibitory concentration (MIC). Then, 5 µL of each sample was deposited on plate count agar (PCA) for 24 h at 37 °C to detect the minimal bactericidal concentration (MBC). Relationship between antibacterial concentration in mg/L and μM used in this work is presented in Appendix A. The data of antibacterial activity of tested cationic CBS dendrimers in μM are presented in Appendix A. 

### 4.4. Analysis of Antibacterial Properties of Dendrimers against Biofilm

#### 4.4.1. Biofilm Formation and Quantification Assay

The protocol previously described by Peppoloni et al. was used to prepare a *P. aeruginosa* CECT 108 biofilm [41]. Bacteria were grown in PCA at 37 °C for 24 h, and then, several colonies were inoculated in TSB at 37 °C and 150 rpm for another 20 h. The inoculum solution was prepared by resuspending *P. aeruginosa* CECT 108 cells in TSB supplemented with 0.4% (wt./wt.) of glucose until an optical density of 0.08–0.1 (10^8^ CFU/mL) at 625 nm. Subsequently, 100 µL of bacterial culture and 100 µL of TSB were added into sterile 96-wells plates to allow the biofilm formation. After 24 h of incubation at 37 °C, biofilm production was checked using a crystal violet assay [42]. Biofilm quantification was carried out by reading the absorbance of a 96-well plate at 630 nm in a microtiter plate reader (Epoch^TM^, BioTek Instruments, Winooski, VT, USA).

#### 4.4.2. Inhibition of *P. aeruginosa* Biofilm Formation

The analysis of cationic CBS dendrimers’ (**4**, **7**, **10**) activity on biofilm formation was carried out as follows. Briefly, 50 µL of *P. aeruginosa* CECT 108 (10^8^ CFU/mL in 2× supplemented with TSB medium) was treated with 50 µL of different doses of each dendrimer (range of initial concentrations from 16 to 1024 mg/L) in a 96-well microtiter plate, for 24 h at 37 °C. Each treatment was evaluated in triplicate. Additionally, all experiments included a positive control (dendrimer-free control) and negative controls (dendrimer without inoculum and culture medium). The minimal biofilm formation inhibitory concentration (MBIC) was determined using a resazurin colorimetric assay, and MBBC, defined as the minimum bactericidal concentration in biofilm, was detected using the drop plate method (Section 4.4.4.).

#### 4.4.3. Antibiofilm Activity Testing

For determining the antibiofilm properties of dendrimers, the supernatant was discarded from the mature biofilm, and cells not adhering to the surface were removed by washing twice with 250 µL of sterile PBS. Then, 100 µL of each dendrimer sample at the concentrations range 16–1024 mg/L was added in triplicate and supplemented with a TSB medium. The plate was incubated at 37 °C for 24 h. In all experiments, positive and negative controls were included. The resazurin colorimetric protocol was followed to determine the minimum biofilm damaging concentration (MBDC) and considered as the lowest concentration that could affect *P. aeruginosa* CECT 108 metabolic activity, while the minimum biofilm eradication concentration (MBEC) was obtained using the drop plate method (Section 4.4.4).

#### 4.4.4. Resazurin Assay and Drop Plate Method

Biofilm cultures after treatment with dendrimers at the concentration range 16–1024 mg/L at 37 °C for 24 h were washed twice with sterile PBS. Then, 100 µL of sterile PBS and 20 µL of resazurin solution were added to each well, and samples were incubated in the dark for 24 h at 37 °C. The reduction of resazurin, in metabolically active cells, was measured by the absorbance at 570 and 600 nm using a microplate reader EpochTM (BioTek Instruments, Winooski, VT, USA). Results obtained using the resazurin assay were confirmed by inoculating 5 µL of each dendrimer-treated culture and controls on plate count agar [33,34].

### 4.5. Scanning Electron Microscopy

Scanning electron microscopy (SEM) was performed to evaluate morphological changes in *P. aeruginosa* cell wall after treatment with dendrimers in TSB at 37 °C for 24 h. The samples for SEM were prepared following the methodology previously described [43,44]. Critical-point drying was carried out with a Polaron CPD7501 critical-point drying system and sputter coated with 200 Å gold–palladium using a Polaron E5400. Scanning electron microscopy was performed at 5–15 kV in a Zeiss DSM 950 SEM.

### 4.6. Laser Interferometry

The diffusion of cationic CBS dendrimers (**4**, **7**) through PAO1 biofilm was determined experimentally using the laser interferometry method [45,46,47]. A biofilm of PAO1 was formed in TSB (tryptic soy broth) medium for 72 h in stationary conditions at 37 °C on a PET membrane with pores of diameter 1 µm as an element of BD Falcon™ Cell Culture Inserts. The laser interferometric system consists of a two-beam Mach–Zehnder interferometer with a He–Ne laser type HN 40P (Carl Zeiss,, Jena, Germany), two cuvettes made with optical glass of high uniformity, a TV-CCD camera and a computer with software for the acquisition and processing of interference images (interferograms). The *P. aeruginosa* POA1 biofilm formed on the PET membrane was placed between the cuvettes. The lower cuvette contained a dendrimer–water solution at (9.06 × 10^−4^ mol/L). The bottom cuvette was filled with pure water. The transport properties of dendrimers through the *P. aeruginosa* PAO1 biofilm were measured using the above laser interferometry system based on the obtained interferograms (Figure 6).

A computer image-processing system, complete with dedicated software, enables mathematical analysis of interferograms shown on the system screen. The interferograms, which appear due to the interference of laser beams, were determined by the refraction coefficient of the solute, which in turn depends on substance concentration. When the solute is uniform, the interference fringes are straight, and they bend when a concentration gradient appears. The basis for the determination of the spatiotemporal substance concentration distribution (e.g., concentration profile), *C*(*x*,*t*), is the proportionality coefficient between changes in substance concentration and the corresponding changes of the solution’s refractive index, *∆n*(*x*,*t*). The correlation between concentration and refractive index of water solutions is refractometrically determined. The concentration profile, *C*(*x*,*t*), is determined by the deviation *d*(*x*,*t*) of the fringes from a straight course. Since the concentration *C*(*x*,*t*) and the refraction coefficient are assumed to be linear [3]:(1)$$C\left(x,t\right)=C_{0}+a∆n=C_{0}+a\frac{\lambda d\left(x,t\right)}{hf},$$
where *C*(*x*,*t*) denotes the concentration of dendrimer at a point situated at the distance *x* from the biofilm–water interface; *C*_0_ is the initial substance concentration; *a* is the proportionality constant between the concentration and the refraction index (a = 1.975 mol/L for the compound **4** and *a* = 0.478 for the compound **7** dendrimers in PBS solution); *λ* is the wavelength of the laser light, 632.8 nm; *h* is the distance between the fringes in the field where they are straight lines and *f* is the thickness of the solution layer in the measurement cuvette. By recording the interferograms over a given time interval (2 min), one can reconstruct the concentration profiles at different times. The interferograms were recorded, and the concentration profiles for each interferogram were reconstructed. Based on concentration profiles, one can determine the transport parameters of substance quantity, such as substance quantity after time *t* (*N*(*t*)):(2)$$N\left(t\right)=S\int_{0}^{\delta }C\left(x,t\right)dx,$$
where *S* is the surface of biofilm (S = 7 × 10^−5^ m^2^), and *δ* is defined as the distance between the biofilm–PBS interface and the point at which the deviation of the interference fringe from its straight-line run amounts to 10% of the fringe thickness.

### 4.7. Fluorescence Microscopy

The *P. aeruginosa* PAO1 mature biofilm was incubated with cationic CBS dendrimers (**4**, **7**) at 9.06 × 10^−4^ mol/L (as in laser interferometry study) for 24 h at 37 °C and stained at room temperature in the dark for 20 min with a mixture of two dyes according to the manufacturer’s protocol (FilmTracer™, Invitrogen, Waltham, USA). The mixtures included two compounds: SYTO^®^
**9**, a green-fluorescent nucleic acid stain, and a red-fluorescent nucleic acid stain, propidium iodide. The PAO1 biofilm was observed with a confocal microscope, Nikon A1R, no longer than 30 min after staining. Interpretation of the microscopic observations was based on the stained cells. Dyes differ both in their spectral characteristics and in their ability to infiltrate healthy bacterial cells. Bacterial cells with green fluorescence are interpreted as alive, whereas bacteria with damaged membranes show red fluorescence.

### 4.8. Antibacterial Activity of Endolysin Combined with Dendrimers

The purified, recombinant endolysin (with the activity of l-alanyl-d-glutamate endopeptidase) derived from the *Klebsiella* KP27 lytic phage was prepared according to methods described previously [22]. Pooled endolysin was dialyzed against phosphate-buffered saline (PBS), pH = 7.4 (1 mL of protein against 10 L of buffer). The concentration of recombinant endolysin was determined fluorimetrically (Qubit^®^ Protein Assay Kit, Molecular Probes, Thermo Fischer Scientific, Waltham, USA). The antibacterial activity of dendrimers was tested using *P. aeruginosa* PAO1 wild-type and its ∆*wbpL* mutant deficient in the biosynthesis of A-band and B-band O-antigens. The antigen-O-lacking mutant was chosen to check the possible effect of this part of LPS on the antibacterial properties of the tested dendrimers. In addition, the dendrimer was combined with phage KP27 endolysin, as a peptidoglycan-degrading enzyme. The antibacterial activity was measured on an exponentially growing *P. aeruginosa* culture (OD_600_) by a spectrometric method using Microplate Reader TECAN Infinite 200 PRO (Tecan Group Ltd., Männedorf, Switzerland). Each dendrimer, alone and in combination with endolysin (5 mmol/L), was added to the bacterial culture, and the bactericidal effect was determined.

## 5. Conclusions

In conclusion, unPEGylated cationic CBS dendrimers are more efficient as a biocide against *P. aeruginosa* than the PEGylated versions. The synergic antibacterial effect was associated with the destabilization of the OM layer by unPEGylated dendrimer and enhanced further by phage-derived endolysin. Finding a good balance in tested dendritic systems and their combination with PG-degrading enzymes, such as endolysin, could be the way toward the future development of biocides against *P. aeruginosa*.

## Figures and Tables

**Figure 1 ijms-23-01873-f001:**
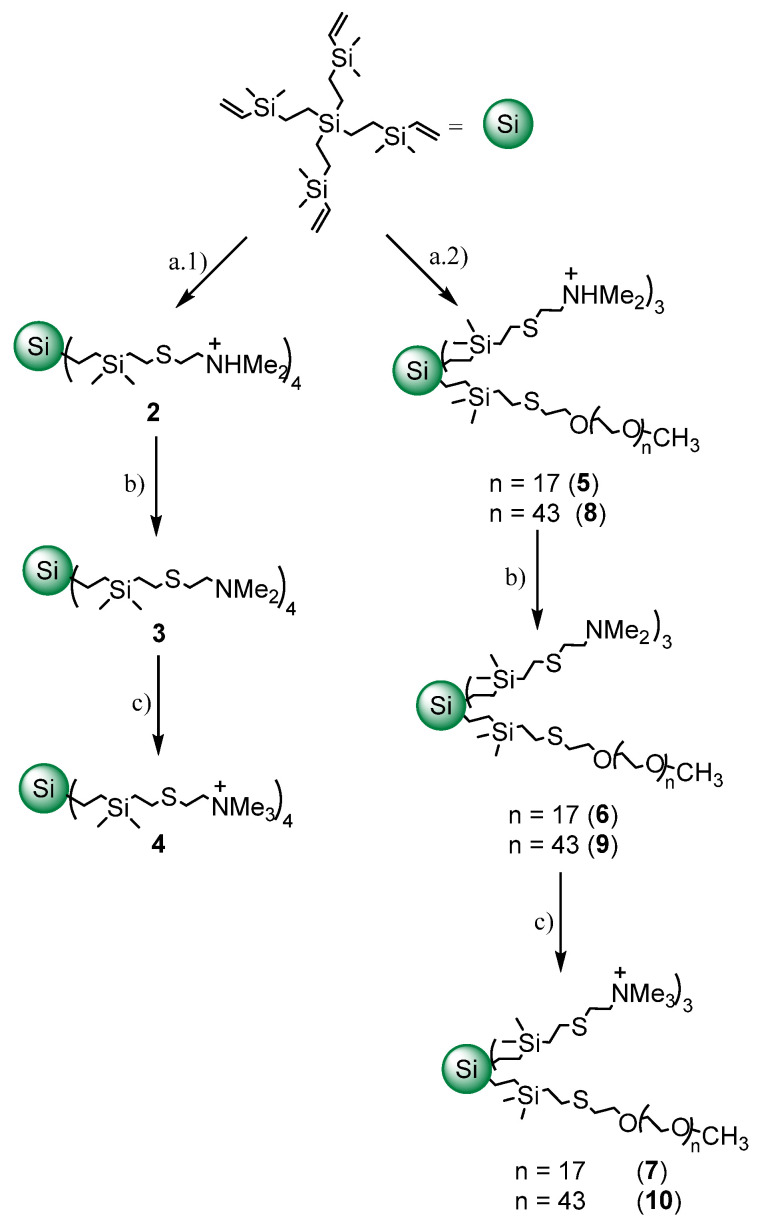
Synthesis of unPEGylated and PEGylated cationic CBS dendrimers. Experimental conditions: (**a.1**) 5% DMPA, hν, THF/MeOH (1:2), HSCH_2_CH_2_NH_2_·HCl (4 equiv.), 4 h; (**a.2**) 5% DMPA, hν, THF/MeOH (1:2), HS-PEG800 or HS-PEG2K (1 equiv.) for 30 min, then HSCH_2_CH_2_NH_2_·HCl (3 equiv.) 4 h; (**b**) excess of NaHCO_3_, distilled H_2_O, 1 h; (**c**) excess MeI, dry THF, 16 h.

**Figure 2 ijms-23-01873-f002:**
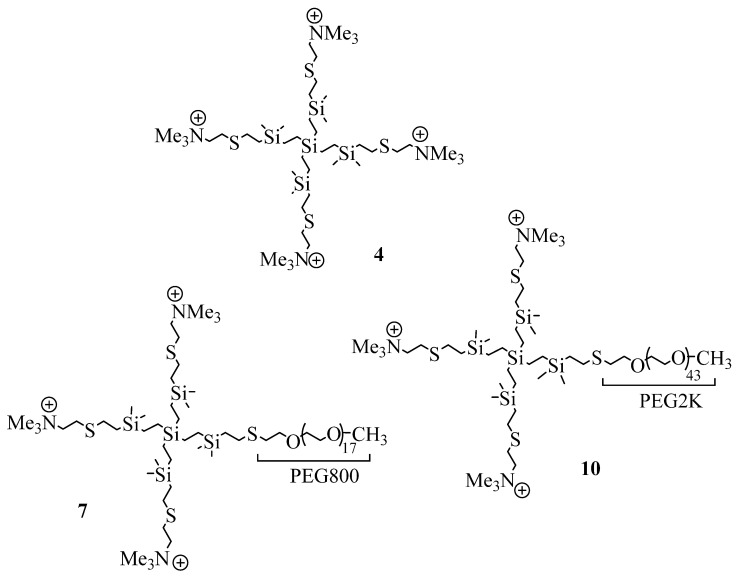
Drawing of cationic CBS dendrimers homo- (**4**) and heterofunctionalized with PEG-800 (**7**) and PEG-2000 (**10**) used in this work.

**Figure 3 ijms-23-01873-f003:**
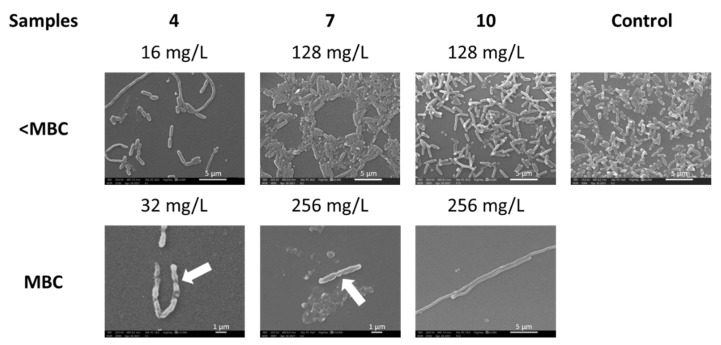
SEM observations of *P. aeruginosa* cell morphology after treatment with cationic CBS dendrimer (**4**) and with pegylated cationic CBS dendrimers (PEG-800 (**7**) or PEG-2000 (**10**)). *P. aeruginosa* cells were treated at each MBC and below. Arrows indicate malformations in the cell wall.

**Figure 4 ijms-23-01873-f004:**
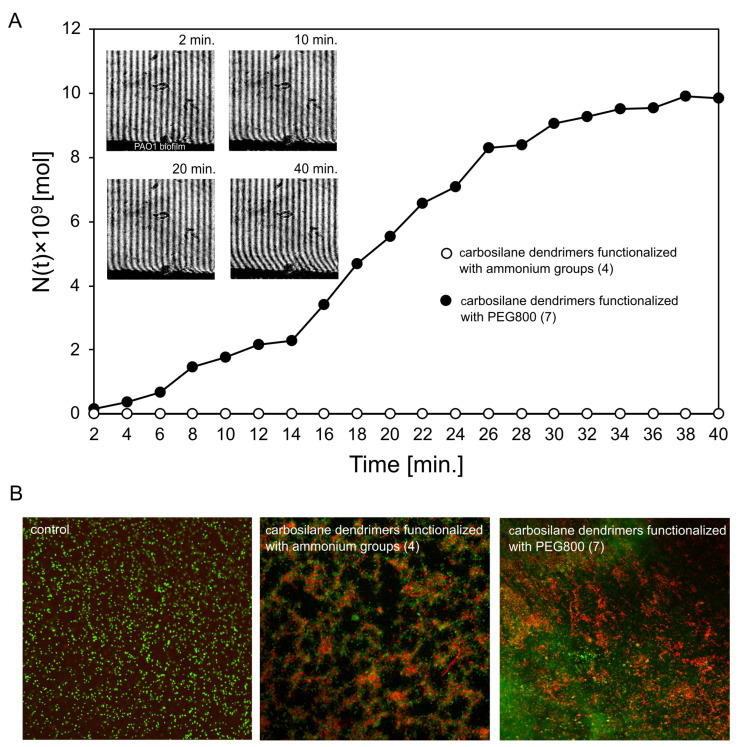
The diffusion of CBS dendrimers (compounds **4** and **7**) through *P. aeruginosa* PAO1 biofilm layer determined by the laser interferometry (**A**). The examples of interferograms for compound **7** are presented above the diffusion curve. (**B**) The fluorescence microscopy images of PAO1 biofilm after treatment with dendrimers (**4** and **7**). Bacterial cells with green fluorescence are interpreted as alive, whereas bacteria with damaged membranes show red fluorescence.

**Figure 5 ijms-23-01873-f005:**
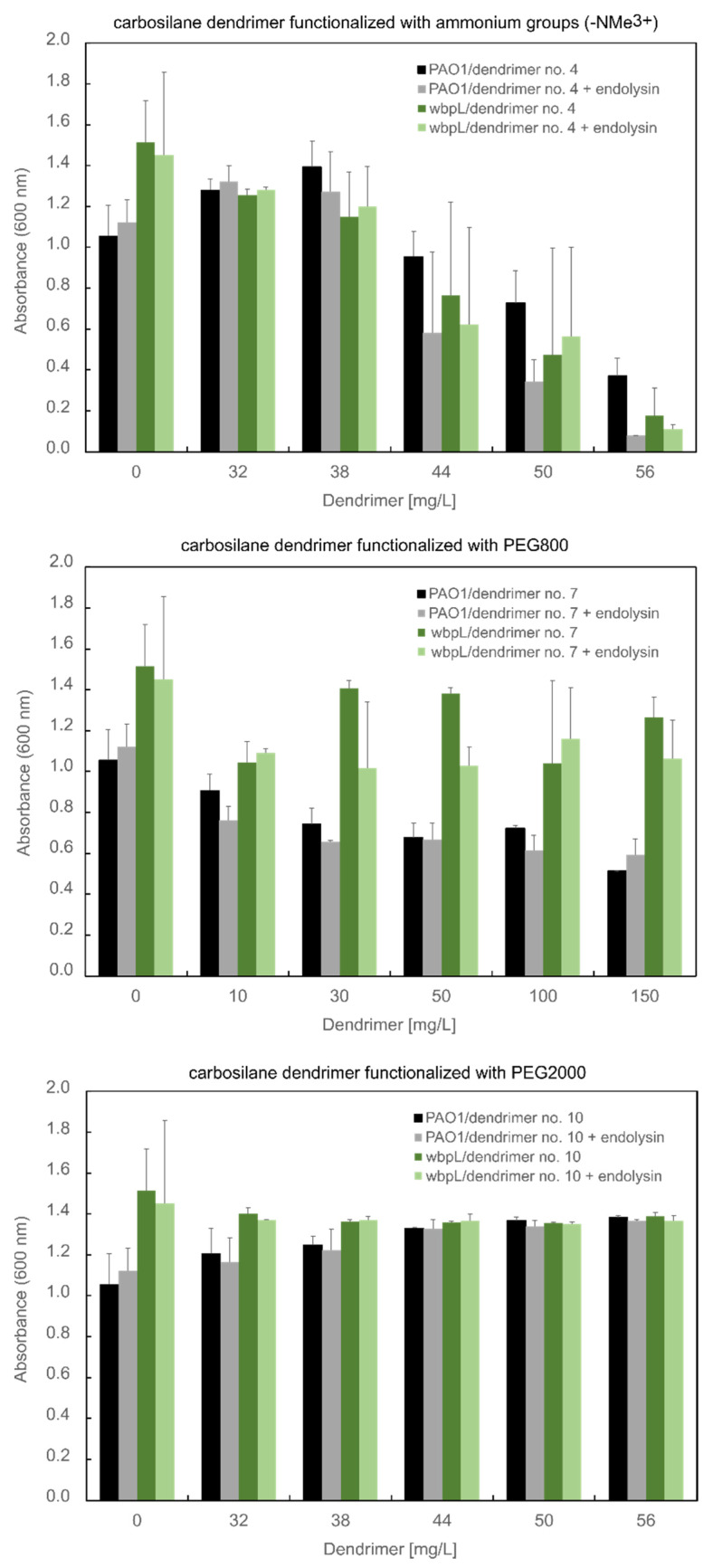
The optical density (OD) at 600 nm of PAO1 wild-type and its ∆*wbpL* mutant after treatment with cationic dendrimers with a constant concentration of KP27 endolysin (5 mM). Results are expressed as mean ± S.D.

**Figure 6 ijms-23-01873-f006:**
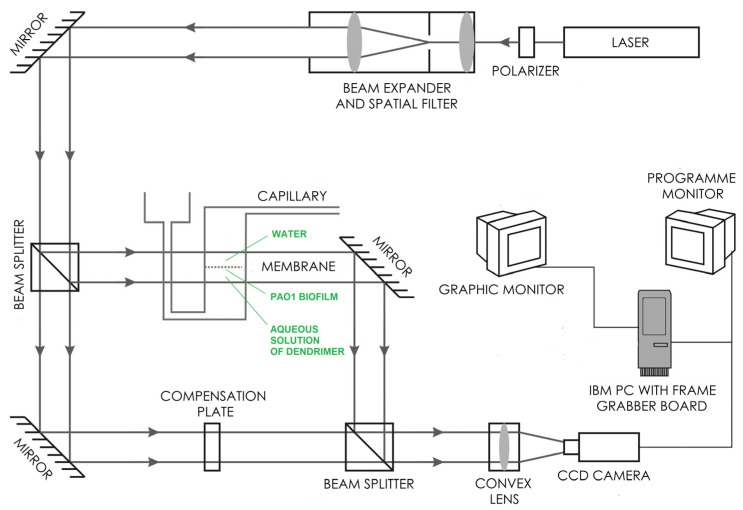
Experimental setup of the interferometric investigations of the dendrimers’ diffusion through a *P. aeruginosa* PAO1 biofilm. The scheme presents a Mach–Zehnder interferometer with a He–Ne laser and a system for the acquisition and processing of interference images. The laser light was spatially filtered and transformed by the beam expander into a parallel beam ca. 80 mm wide and then split into two beams. The first beam passed through the investigated membrane system parallel to the membrane surface, while the second passed directly through the compensation plate to the light detection system. As a consequence of the superimposition of these beams, respective interference images were generated. The images depend on the refraction coefficient of the solute, which in turn depends on the substance concentration. When the solute is uniform, the interference fringes are straight, and they bend when a concentration gradient appears.

**Table 1 ijms-23-01873-t001:** Antibacterial activity of cationic CBS dendrimer (**4**) and with pegylated cationic CBS dendrimer PEG-800 (**7**) or PEG-2000 (**10**) against *P. aeruginosa* planktonic cells and biofilm. Data are given in mg/L.

Dendrimer (Compound No.)	Planktonic Cells		Biofilm			
			Preventing Biofilm Formation		Removing Biofilm	
	MIC	MBC *	MBIC	MBBC *	MBDC	MBEC *
4	32	32	64	64–128	128	512
7	256	256	512	512–1024	512–1024	1024–>1024
10	256	256	>1024	>1024	>1024	>1024

* Drop plate method. Planktonic cells: minimum inhibitory concentration (MIC) and minimum bactericidal concentration (MBC). Biofilm: minimum biofilm inhibitory concentration (MBIC), minimum bactericidal concentration in biofilm (MBBC), minimum biofilm damaging concentration (MBDC) and minimum biofilm eradication concentration (MBEC).

## Data Availability

Not applicable.

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
