# Peer review of "The Antibacterial Effect of PEGylated Carbosilane Dendrimers on P. aeruginosa Alone and in Combination with Phage-Derived Endolysin"

_ijms, 2022, doi:10.3390/ijms23031873_

Round 1
Reviewer 1 Report
The emergence and spread of antibiotic-resistant bacteria is a serious global public health problem that affects the effectiveness of medical treatments of many infectious diseases. The situation is urgent with so-called ESKAPE pathogens, with Pseudomonas aeruginosa being a prominent member with a strong tendency for biofilm formation.
The authors evaluated the antibacterial activity of positively charged carbosilane dendrimers with ammonium groups unmodified and modified with polyethylene glycol against gram-negative bacterium Pseudomonas aeruginosa. They found that the presence of polyethylene glycol reduced the antibacterial activity of dendrimers. However, they found that the activity can be improved by combining dendrimers and KP27 endolysin.
The issues concerning the mechanism of antibacterial activity and cytotoxicity of dendrimers alone and PEGylated should be addressed. In the paper on a similar subject, where amino-terminated PAMAM dendrimers were used, the Authors found that amino-terminated PAMAM is toxic to HCECs cells at concentration 10 micrograms/ml (Biomacromolecules 8 (2007) 1807-1811). Another question is the mechanism of action. It was found that, at high concentration, PAMAM can form nanopores in eukaryotic membranes (Bioconjugate Chem. 11 (2000) 805-814. We can hypothesize that the exact mechanism is involved in the case of carbosilane dendrimers with ammonium groups and the target bacterial cytoplasmic membrane, which eventually lead to cell death. In this respect applying a lipid membrane screen to identify molecular targets for dendrimers would be helpful (see Bio Protocol 7 (2017) e2427, doi 10.21769/BioProtoc.2427). The antibacterial effect of dendrimers on P. aeruginosa mutant lacking lipopolysaccharide O-chain is also interesting worthy of commenting (Fig. 5). It suggests interactions of dendrimers with the carbohydrate moiety.
Table 1. Please provide the units for parameters of antibacterial activity
Subchapter 2.5 starts with “Endolysins are a type of enzyme that can be a biocide against Gram-positive bacteria”. Endolysins are enzymes produced by phages infecting both gram-positive and gram-negative bacteria. They can lyse both types of bacteria depending on the enzyme origin and properties. I think the Authors wanted to stress the difficulty of lysis of gram-negative bacteria (lysis from without or external lysis) due to the presence of the outer membrane.
Fig. 3. SEM observations of Pseudomonas aeruginosa morphology. The images are stunning in the case of compound no. 10 at concentration 256 mg/ml. We can observe the formation of long filaments. This finding suggests that dendrimers induce a systemic response in bacterial cells. Please comment on this. In Pseudomonas putida, filamentation occurs due to the lack of septation during the cell division process. It results in the formation of elongated bacteria, which is typically a consequence of DNA damage or envelope stress (BMC Microbiology 12 92012) 282). In P. aeruginosa, filamentation was observed under anaerobic growth (Plos One 6 (2011) e16105).
Have the Authors analyzed the morphology of bacterial cells treated with dendrimers and endolysin?
The information on KP27 endolysin of phage vB_KpnM_KP27 is scarce in the text. The enzyme shows L-alanyl-D-glutamate endopeptidase activity (Appl. Microbiol. Biotechnol. 101 (2017) 673-684).
Discussion. It would be appropriate to overview recent findings regarding factors that facilitate outer membrane permeabilization. In this respect, an interesting discovery was made recently in the case of a viral homolog of PGRP proteins (Scientific Reports 9 (2019) 1261). It was proved for Thermus phage Ts2631 endolysin that positively charged 20-aa N-terminal extension is critical for exerting antibacterial activity. It allows the enzyme to pass through the outer membrane and interact with murein. The same group showed the effectiveness of the Ts2631 enzyme in combating multidrug-resistant gram-negative pathogens of Acinetobacter baumannii and Pseudomonas aeruginosa (Viruses 11 (2019) 657) . Another example of this kind of protein is LysC of Clostridium intestinale which uses N-terminal extension to form nanopores in the cytoplasmic membrane (IJMS 21 (2020) 4894).
Author Response
27 February 2022
Thank you for your comments concerning our manuscript. We need to stress that your comments allowed us to improve the quality of the manuscript. Enclosed are our answers to comments and an outline of all changes we have made “in green” on the manuscript.
- The issues concerning the mechanism of antibacterial activity and cytotoxicity of dendrimers alone and PEGylated should be addressed. In the paper on a similar subject, where amino-terminated PAMAM dendrimers were used, the Authors found that amino-terminated PAMAM is toxic to HCECs cells at concentration 10 micrograms/ml (Biomacromolecules 8 (2007) 1807-1811). Another question is the mechanism of action. It was found that, at high concentration, PAMAM can form nanopores in eukaryotic membranes (Bioconjugate Chem. 11 (2000) 805-814. We can hypothesize that the exact mechanism is involved in the case of carbosilane dendrimers with ammonium groups and the target bacterial cytoplasmic membrane, which eventually lead to cell death. In this respect applying a lipid membrane screen to identify molecular targets for dendrimers would be helpful (see Bio Protocol 7 (2017) e2427, doi 10.21769/BioProtoc.2427).
This information has been added to the Discussion section.
- The antibacterial effect of dendrimers on P. aeruginosa mutant lacking lipopolysaccharide O-chain is also interesting worthy of commenting (Fig. 5). It suggests interactions of dendrimers with the carbohydrate moiety.
Information describing the interaction of the dendrimers with the LPS has now been added to the text.
- Table 1. Please provide the units for parameters of antibacterial activity.
The units were added to the Table 1 legend.
- Subchapter 2.5 starts with “Endolysins are a type of enzyme that can be a biocide against Gram-positive bacteria”. Endolysins are enzymes produced by phages infecting both gram-positive and gram-negative bacteria. They can lyse both types of bacteria depending on the enzyme origin and properties. I think the Authors wanted to stress the difficulty of lysis of gram-negative bacteria (lysis from without or external lysis) due to the presence of the outer membrane.
Thank you for your important attention. The manuscript has been corrected. Please, see the start of Subchapter 2.5.
- Fig. 3. SEM observations of Pseudomonas aeruginosa morphology. The images are stunning in the case of compound no. 10 at concentration 256 mg/ml. We can observe the formation of long filaments. This finding suggests that dendrimers induce a systemic response in bacterial cells. Please comment on this. In Pseudomonas putida, filamentation occurs due to the lack of septation during the cell division process. It results in the formation of elongated bacteria, which is typically a consequence of DNA damage or envelope stress (BMC Microbiology 12 92012) 282). In P. aeruginosa, filamentation was observed under anaerobic growth (Plos One 6 (2011) e16105)
It has been added to the Discussion section.
- Have the Authors analyzed the morphology of bacterial cells treated with dendrimers and endolysin?
We did not analyze the morphology of bacterial cells in the presence of dendrimers and endolysin. We focused rather on dendrimers as membrane permeabilizing agents for endolysin.
- The information on KP27 endolysin of phage vB_KpnM_KP27 is scarce in the text. The enzyme shows L-alanyl-D-glutamate endopeptidase activity (Appl. Microbiol. Biotechnol. 101 (2017) 673-684).
This important information has been added, please see subchapters 2.5 and 4.8
- Discussion. It would be appropriate to overview recent findings regarding factors that facilitate outer membrane permeabilization. In this respect, an interesting discovery was made recently in the case of a viral homolog of PGRP proteins (Scientific Reports 9 (2019) 1261). It was proved for Thermus phage Ts2631 endolysin that positively charged 20-aa N-terminal extension is critical for exerting antibacterial activity. It allows the enzyme to pass through the outer membrane and interact with murein. The same group showed the effectiveness of the Ts2631 enzyme in combating multidrug-resistant gram-negative pathogens of Acinetobacter baumannii and Pseudomonas aeruginosa (Viruses 11 (2019) 657). Another example of this kind of protein is LysC of Clostridium intestinale which uses N-terminal extension to form nanopores in the cytoplasmic membrane (IJMS 21 (2020) 4894).
Thank you for your important comment. Recent findings regarding endolysins applicability towards Gram-negative pathogens have been added to the discussion section in the manuscript body.
We hope that the major revision, corrected version of our manuscript will be acceptable for publication in the International Journal of Molecular Sciences.
Sincerely yours
Javier Sánchez-Nieves
Michał Arabski
corresponding authors
Reviewer 2 Report
On request of IJMS, I have revised the manuscript titled “The antibacterial effect of PEGylated carbosilane dendrimers on P. aeruginosa alone and in the combination with phage-derived endolysin” by Sara Quintana-Sanchez et al.
Aiming at developing new microbicide compounds especially functioning against biofilm-forming bacteria, in this work unmodified and PEGylated cationic carbosilane (CBS) dendrimers were synthetized and characterized by spectroscopic techniques and elemental analyses. The antibacterial effects of the prepared materials, alone and in combination with endolysin, were subsequently evaluated against planktonic and biofilm-forming P. aeruginosa with appealing outcomes.
General Comments
The present work concerns a very interesting topic and reports appealing results both in the field of synthetic cationic dendrimers and in that of microbiology. The manuscript is well-done and well-written. The English language is fine. The presentation of the abstract provides readers with a useful detail for framing the topic. Additionally, even if the authors have already explored and reported the synergistic antimicrobial effects provided by the combination of cationic dendrimers with endolysin against P. aeruginosa, in this case have synthetized and tested different types of dendrimers and organized their study in an original way with respect to their previous work. As an organic chemist, I have appreciated particularly the experimental section concerning the synthesis and characterization of the carbosilane dendrimers, which, although not complete, has been well exposed.
However, the work can be further improved, by expanding the introduction, performing additional experiments, and providing additional data. Finally, some minor issues must be addressed before the herein manuscript is suitable for publication on IJMS.
Introduction.
Line 60. Here the authors have not considered the polyester-based dendrimers, which instead, have been and are extensively studied mainly because of their high level of biodegradability and low toxicity. Please, see:
Macromolecules 1998, 31, 4061–4068
Org. Commun. 2017, 10, 144–177
Macromol. Res. 2017, 25, 1172–1186
Drug Deliv. Transl. Res. 2019, 10, 259–270
Line 62. Here the authors have not considered to cite recent works concerning the antibacterial properties of cationic dendrimers. Please see at:
Nanomaterials 2020, 10, 2022.
Int. J. Mol. Sci. 2021, 22, 7274. https://doi.org/10.3390/ijms22147274
Pharmaceutics 2021, 13, 1411. https://doi.org/10.3390/pharmaceutics13091411 replace “based in” with “based on”.
The authors should provide the PDF files of all 1H, 13C NMR spectra and those of mass spectra as Supplementary Materials.
The authors should provide the MW of all intermediate and of final dendrimers. If the authors have not MALDI-TOF, can estimate the required MW by 1H NMR spectra using the values of properly selected integrals. Once obtained, the MW values should be compared with those obtainable from the formula of the compounds. The related discussion should be inserted.
Using the MW of final dendrimers, the authors should provide the MICs and other microbiological data expressing them in micromolar concentrations.
The authors should investigate the size, morphology, PDI and zeta-potential of dendrimer particles by SEM and DLS analysis. A discussion of results should be inserted.
Minor issues
In both the title and the sections and sub-sections titles, al words with first letters as capital letters.
All Figure captions should be in a justified format.
Please, check all the abbreviations and specify them at the first mention.
Line 33. There is something wrong in this sentence. Please, check and correct.
Despite, the good quality of the original work it should be improved according to the present report.
So, knowing the times allowed by IJMS for revision, since I think that the necessary revisions will require authors considerable effort and time, I decided to ask for major revision, so that authors will be allowed to have sufficient time. I will be glad to reconsider the present work once the above-mentioned revision will be addressed.
Author Response
27 February 2022
Thank you for your comments concerning our manuscript. We need to stress that your comments allowed us to improve the quality of the manuscript. Enclosed are our answers to comments and an outline of all changes we have made “in green” on the manuscript.
- Line 60. Here the authors have not considered the polyester-based dendrimers, which instead, have been and are extensively studied mainly because of their high level of biodegradability and low toxicity. Please, see: Macromolecules 1998, 31, 4061–4068, Org. Commun. 2017, 10, 144–177, Macromol. Res. 2017, 25, 1172–1186, Drug Deliv. Transl. Res. 2019, 10, 259–270
There are several types of dendrimers, which are usually classified by their framework. As the referee comments, polyester dendrimers are also another important type of dendrimers. We have included this important type of dendrimers in the list, but we have changed the references introducing some reviews that discuss the wide variety of dendrimers.
Before change: Nowadays, different dendritic systems are being developed with a variety of frameworks, such as PAMAM (poly(amidoamine)) [11], PPI (poly(propylene imine)) [12], carbosilane (CBS) [13], phosphorus [14] or polyether dendrimers [15].
After change: Nowadays, different dendritic systems are being developed with a variety of frameworks [REFS], such as PAMAM (poly(amidoamine)), PPI (poly(propylene imine)), carbosilane (CBS), phosphorus, polyether or polyester dendrimers.
For the new REFS:
1. Tomalia, D.A.; Baker, H.; Dewald, J.; Hall, M.; Kallos, G.; Martin, S.; Roeck, J.; Ryder, J.; Smith, P. A new class of polymers : Starburst-Dendritic. Polym. J. 1985, 17, 117–132.
2. Vögtle, F.; Richardt, G.; Werner, N. Dendrimer Chemistry: Concepts, Syntheses, Properties, Applications; Wiley-VCH, 2009; ISBN 978-3-527-62696-0.
3. Malkoch, M., García Gallego, S. Dendrimer Chemistry: Synthetic Approaches Towards Complex Architectures; The Royal Society of Chemistry, 2020; ISBN 978-1-78801-132-7.
- Line 62. Here the authors have not considered to cite recent works concerning the antibacterial properties of cationic dendrimers. Please see at: Nanomaterials 2020, 10, 2022., Int. J. Mol. Sci. 2021, 22, 7274. https://doi.org/10.3390/ijms22147274, Pharmaceutics 2021, 13, 1411. https://doi.org/10.3390/pharmaceutics13091411.
We have actualized the references of this sentence as follows:
1. Ortega, P.; Sánchez-Nieves, J.; Cano, J.; Gómez, R.; de la Mata, F.J. Poly (carbosilane) dendrimers and other silicon-containing dendrimers. In Dendrimer Chemistry: Synthetic Approaches Towards Complex. Architectures; Edited by Michael Malkoch and Sandra García Gallego; The Royal Society of Chemistry 2020, 114-145.
2. Chen, A.; Karanastasis, A.; Casey, K.R.; Necelis, M.; Carone, B.R.; Caputo, G.A.; Palermo, E.F. Cationic Molecular Umbrellas as Antibacterial Agents with Remarkable Cell-Type Selectivity. ACS Appl. Mater. Interfaces 2020, 12, 21270–21282, doi:10.1021/acsami.9b19076.
3. Alfei, S.; Schito, A.M. From nanobiotechnology, positively charged biomimetic dendrimers as novel antibacterial agents: A review. Nanomaterials 2020, 10, 1–50, doi:10.3390/nano10102022.
3.
3.1. The authors should provide the PDF files of all 1H, 13C NMR spectra and those of mass spectra as Supplementary Materials.
A Supporting Information containing the information about NMR data has been attached to the manuscript.
3.2. The authors should provide the MW of all intermediate and of final dendrimers. If the authors have not MALDI-TOF, can estimate the required MW by 1H NMR spectra using the values of properly selected integrals. Once obtained, the MW values should be compared with those obtainable from the formula of the compounds. The related discussion should be inserted.
We have tried to characterize the dendrimers by Mass Spectrometry but we have been unsuccessful. This technique, ESI or MALDI-TOF, sometimes failed with CBS dendrimers due to the presence of the aliphatic chains, which exhibit low ionization without fragmentation (see e.g. Anal. Chem. 2007, 79, 1639-1645).
For some of the intermediates, as the compounds with –NMe2H+ groups (compound 8), we have not included the molecular weight because we have not purified the compounds, since these compounds have been used for the next steps after checking that the reaction has been finished. However, compounds obtained from then were properly purified and characterized.
3.3. Using the MW of final dendrimers, the authors should provide the MICs and other microbiological data expressing them in micromolar concentrations.
The standard unit for concentration in microbiology is mg/l (or ppm). However, to better understand the behaviour of these compounds we have given this information also in molar concentration, as the referee suggests. These data are collected in tables in the supporting information. Moreover, a paragraph comparing activity in molar concentration has been also added in the discussion section.
3.4. The authors should investigate the size, morphology, PDI and zeta-potential of dendrimer particles by SEM and DLS analysis. A discussion of results should be inserted.
The size of these dendrimers is too small to be detected by Potential Zeta, DLS or electron microscopy even for the dendrimer with the PEG2k unit. They resemble more small oligomers. This fact has been observed usually for dendrimers of low generation as presented in this work. The molecular weight of the main dendrimers in this article is below 2k. Taking into account that the dendrimers are cationic, that is, their weight is smaller without the counter ion, these techniques are not useful for them. For example, the radius of bigger dendrimers, with 16 functional groups is below 2 nm (Dalton Trans., 2012, 41, 12733).
- In both the title and the sections and sub-sections titles, al words with first letters as capital letters.
This change has been done.
- All Figure captions should be in a justified format.
This change has been done.
- Please, check all the abbreviations and specify them at the first mention.
The manuscript has been revised to amend this problem.
- Line 33. There is something wrong in this sentence. Please, check and correct.
This sentence has been modified in the Abstract section.
We hope that the major revision, corrected version of our manuscript will be acceptable for publication in the International Journal of Molecular Sciences.
Sincerely yours
Javier Sánchez-Nieves
Michał Arabski
corresponding authors

Round 2
Reviewer 2 Report
Although the authors have not completely satisfied my requests and I do not agree with the fact that the prepared particles are too small for a DLS analysis, I believe that enough work has been done to improve the original manuscript which can now be published on IJMS.